# Prevalence of Antimicrobial Resistance in *Klebsiella pneumoniae*, *Enterobacter cloacae*, and *Escherichia coli* Isolates among Stillbirths and Deceased Under-Five Children in Sierra Leone: Data from the Child Health and Mortality Prevention Surveillance Sites from 2019 to 2022

**DOI:** 10.3390/microorganisms12081657

**Published:** 2024-08-13

**Authors:** Julius Ojulong, Gebrekrstos N. Gebru, Babatunde Duduyemi, Edwin Gbenda, Mohamed L. Janneh, Jack Sharty, Leonel Monteiro, Dickens Kowuor, Soter Ameh, Ikechukwu U. Ogbuanu

**Affiliations:** 1CHAMPS Program Office, Emory Global Health Institute, Emory University, Atlanta, GA 30322, USA; egbenda@emory.edu (E.G.); mjanneh@emory.edu (M.L.J.); jsharty@emory.edu (J.S.); dkowuor@emory.edu (D.K.); sameh@emory.edu (S.A.); iogbuanu@emory.edu (I.U.O.); 2Sierra Leone Field Epidemiology Training Program, Africa Field Epidemiology Network, Freetown 232, Sierra Leone; ggebru@afenet.net; 3College of Medicine and Allied Health Sciences, University of Sierra Leone, Freetown 232, Sierra Leone; tundeduduyemi@gmail.com; 4Independent Researcher, Maputo 1102, Mozambique; lmonteiro1438@gmail.com; 5Department of Community Medicine, Faculty of Clinical Sciences, University of Calabar, Calabar 540281, Nigeria; 6Bernard Lown Scholars Program in Cardiovascular Health, Department of Global Health and Population, Harvard T. H. Chan School of Public Health, Boston, MA 02115, USA; 7Hubert Department of Global Health, Rollins School of Public Health, Atlanta, GA 30322, USA

**Keywords:** *Klebsiella pneumoniae*, *Enterobacter cloacae*, *Escherichia coli*, CHAMPS, extended spectrum beta-lactamase, Sierra Leone

## Abstract

*Klebsiella pneumoniae*, *Escherichia coli*, and *Enterobacter cloacae* are associated with most nosocomial infections worldwide. Although gaps remain in the knowledge of their susceptibility patterns, these are in antimicrobial stewardship. This study aimed to describe antimicrobial susceptibility profiles of the above organisms isolated from postmortem blood from stillbirths and under-five children enrolled in the Child Health and Mortality Prevention Surveillance (CHAMPS) program in Sierra Leone. This was a surveillance study of bacteria isolates from postmortem blood cultures taken within 24 h of death from stillbirths and children aged 0–59 months between March 2019 and February 2022. This was followed by identification and antibiotic sensitivity testing using Becton Dickinson Phoenix M50 (USA). Descriptive analysis was used to characterize the isolates and their antimicrobial susceptibility patterns. Of 367 isolates, *K. pneumoniae* was the most frequently isolated organism (*n* = 152; 41.4%), followed by *E. coli* (*n* = 40; 10.9%) and *E. cloacae* (*n* = 35; 9.5%). Using BACTEC™ FX 40 (Franklin Lakes, NJ, USA), 367 isolates were identified from blood using bacteriological methods. Extended spectrum beta-lactamase (ESBL) was observed in 143 (94.1%) of *K. pneumoniae* isolates and 27 (65.5%) of *E. coli* isolates. Carbapenem-resistant organisms (CRO) were seen in 31 (20.4%) of *K. pneumoniae* and 5 (12.5%) of *E. coli* isolates. A multidrug resistance (MDR) pattern was most prevalent in *E.cloacae* (33/35; 94.3%), followed by *K. pneumoniae* (138/152; 90.8%). Our study showed a high prevalence of multidrug resistance among bacterial isolates in the catchment areas under surveillance by the CHAMPS sites in Sierra Leone.

## 1. Introduction

Antimicrobial resistance (AMR) in Gram-positive and Gram-negative bacteria has increased recently. Globally, AMR in a wide range of infectious agents is a growing public health threat [1]. The global proliferation of pathogens that cause common diseases and are resistant to antimicrobial agents is particularly concerning [1]. One of the main drivers behind the development of AMR is the misuse and overuse of antibacterial agents, which render even the most effective drugs ineffective [2,3]. AMR is an increasing problem in both hospital and community settings, associated with the successful penetration and spread of MDR strains from nosocomial settings [4].

Multidrug resistance (MDR) is a phenomenon where microorganisms become resistant to at least one antimicrobial drug in three or more antimicrobial categories [5]. Infections caused by MDR pathogens often need to be treated with more expensive and/or more toxic antibiotics [6,7], which may also affect patient outcomes. The financial burden of AMR is also considerable. One infection caused by an AMR pathogen (e.g., Methicillin-Resistant *Staphylococcus aureus* bloodstream infection) can cost up to USD 20,000 [8].

In the management of sepsis, especially in neonatal intensive care units worldwide, the recommended first-line treatment for both early and late onset neonatal sepsis is a beta-lactam antibiotic (e.g., ampicillin and penicillin) combined with an aminoglycoside (e.g., gentamicin). However, the local evidence in Sierra Leone on the clinical effects of the commonly used antibiotic regimens for sepsis showed that these had limited effectiveness, hence the widespread use of third generation cephalosporins and carbapenems [2,3].

First-line antibiotics (penicillin/ampicillin and gentamicin) have been the mainstay of treatment for serious infections caused by *Enterobacteriaceae*, but efficacy has been eroded by the widespread acquisition of resistance enzymes, such as extended-spectrum beta-lactamases (ESBL) and carbapenemases, which mediate the respective resistance to these critical drugs [9]. ESBL and carbapenem-resistant organisms (CRO) have emerged as an important therapeutic challenge [10,11].

The World Health Organization has listed the following among the priority pathogens against which new antimicrobial agents are urgently needed [10,11]: Acinetobacter baumannii, carbapenem-resistant; Enterobacterales, third-generation cephalosporin-resistant; Enterobacterales, carbapenem-resistant and *Mycobacterium tuberculosis*, rifampicin-resistant. The bulk of hospital-acquired infections are caused by *K. pneumoniae*, *E. coli*, and *Enterobacter cloacae* bacteria, which have the ability to “escape” the biocidal effects of antibiotics [10]. Therefore, these three organisms are in the spotlight of antibiotic stewardship programs and are the most commonly isolated at the Special Baby Care Unit in Makeni Regional Hospital, Sierra Leone (unpublished report). According to a 2015 WHO report [12], they were also the most common to exhibit MDR. Furthermore, *E. coli* and *K. pneumoniae* are the most common and emerging bacteria reported as etiologic agents of premature and neonatal sepsis and pneumonia [12,13,14].

Numerous studies have shown a link between infections by AMR bacteria and unfavorable patient outcomes, including lengthier hospital admissions, increased morbidity, and mortality [15,16,17]. Antibiotic treatments initiated in the emergency departments are often empirical. Comprehensive knowledge of the burden and associated outcomes of infections caused by MDR pathogens is still lacking, especially in West Africa. The knowledge of the local susceptibility patterns of these bacteria is lacking as shown by indiscriminate use of antibiotics and empirical prescriptions by clinicians which are crucial for antimicrobial stewardship programs. Therefore, this study aimed to describe the antimicrobial susceptibility pattern of *K. pneumoniae*, *E. coli*, and *E. cloacae* in Sierra Leone. This knowledge will support the Ministries of Health in similar countries to design targeted approaches to improve antimicrobial stewardship and the impact of the empirical management of childhood infections in LMICs.

## 2. Methods

### 2.1. Study Design

This is a surveillance study of bacteria isolates from postmortem blood cultures taken within 24 h of death from two sites in the Bombali and Bo Districts of Sierra Leone.

### 2.2. Study Site and Settings

The Child Health and Mortality Prevention Surveillance Network (CHAMPS) is a global surveillance network that generates, collects, analyzes, and shares data to reduce child mortality in regions where it is highest—Sierra Leone, Mali, Nigeria, Ethiopia, Kenya, Mozambique, and South Africa in Sub-Saharan Africa; and Pakistan and Bangladesh in Asia. The first CHAMPS site in Sierra Leone is located in Bombali Shebora and Siari chiefdoms in the Bombali District (Northern Province) (Figure 1 below). The Makeni site includes parts of Makeni City (population 125,970) as well as peri-urban and rural areas (population 36,413), resulting in a total site population of 161,383 [18]. The CHAMPS catchment area makes up approximately 25% of the total district population. The estimated population of children under five years of age under surveillance in the catchment area is approximately 24,000, with an approximate under-five mortality rate of 119/1000 live births, a crude birth rate of 32.4 livebirths per 1000 population, and a fertility rate of 4.6/1000 women [18,19].

The second CHAMPS site in Sierra Leone is located in the Bo District in the chiefdoms of Kakua and Tikonko (Southern Province) and including Bo City, with a catchment population of 278,649, a crude birth rate of 32.4 livebirths per 1000 population, an under five years of age mortality rate of 38/1000 live births, and a fertility rate of 4.2/1000 women [18,19].

### 2.3. Study Population and Sample Collection

The isolates are from postmortem blood collected from stillbirths, neonates, and children up to 59 months in two CHAMPS sites, as described elsewhere [20,21]. Blood samples were obtained from the subclavian artery within 24 h of death.

### 2.4. Inclusion Criteria

The criteria for inclusion in the study were as follows: residence in the catchment area within the last four months, a confirmed stillbirth or less than 60 months of age, availability of the body for sample collection, and time between death and eligibility screening/consent being less than or equal to 24 h for an unrefrigerated body or 72 h if the body was refrigerated soon after death.

### 2.5. Exclusion Criteria

The criteria for exclusion in the study were as follows: the body was treated using formalin, or death caused by trauma, drowning, or poisoning.

### 2.6. Culture and Susceptibility Testing

*K. pneumoniae*, *E. coli*, and *E. cloacae* were isolated between February 2019 and February 2022. Blood culture was completed using BACTEC™ FX 40 (Franklin Lakes, NJ, USA), and isolates were identified using the classical bacteriology Gram staining method followed by automated identification and antibiotic sensitivity testing using the Becton Dickinson Phoenix M50 instrument (Franklin Lakes, NJ, USA). Antimicrobial susceptibility trends were ascertained from minimum inhibitory concentrations and interpreted using Clinical and Laboratory Standards Institute guidelines [22]. Any organism showing intermediate resistance was taken as resistant.

The resistance analysis of the *K. pneumoniae*, *E. coli*, and *Enterobacter* spp. isolates also included assessments for extended spectrum beta-lactamase patterns (resistance to penicillins and cephalosporins); carbapenemase resistance (resistance to carbapenems); and multidrug resistance [MDR] (non-susceptibility to at least one agent from at least three antimicrobial categories) [5]. The following antibiotic classes were tested to assign MDR patterns: penicillins (penicillin, amoxicillin–clavulanate); cephalosporins (cefepime, cefotaxime, ceftazidime, cefuroxime, piperacillin–tazobactam); carbapenems (ertapenem, imipenem, meropenem); fluoroquinolones (ciprofloxacin, levofloxacin); aminoglycosides (amikacin, gentamicin, tobramycin); tetracyclines (tetracycline, doxycycline); and sulphonamides (sulfamethoxazole–trimethoprim).

### 2.7. Data Analysis

Descriptive analysis was used to identify the antimicrobial susceptibility patterns among the identified isolates. We assessed the prevalence of different patterns using proportions. Intersection analysis was used to plot the resistance patterns of selected relevant organisms as previously described [23]. Analysis was performed using R 4.2.1.

### 2.8. Ethics

Ethical approval was obtained from the Sierra Leone Ministry of Health and Sanitation. Confidentiality and data security were maintained throughout the analysis and review of the study results.

## 3. Results

From March 2019 to February 2022, 367 organisms were isolated from samples collected from 426 deaths recorded in the CHAMPS study in Sierra Leone. *Klebsiella pneumoniae* accounted for 152 (41.4%) of all isolates, *Escherichia coli* for 40 (10.9%), and *Enterobacter cloacae* for 33 (9.5%) (Table 1).

**Table 1 microorganisms-12-01657-t001:** Isolated organisms in CHAMPS sites, Sierra Leone 2019–2022.

Characteristic	N = 367 ^1^
Isolated organisms	
*Klebsiella pneumonia*	152 (41.4%)
*Escherichia coli*	40 (10.9%)
*Enterobacter cloacae*	35 (9.5%)
Other *	140 (38.1%)

^1^ *n* (%). * *Burkholderia cepacia* complex, *Klebsiella aerogenes*, *Klebsiella oxytoca*, *Acinetobacter baumannii*, *Staphylococcus haemolyticus*, *Staphylococcus epidermidis*, *Enterococcus faecalis*, *Staphylococcus aureus*, *Salmonella species*, *Pseudomonas aeruginosa*, *Serratia marcescens*, *Aeromonas hydrophila*.

ESBL was identified in 243 (66.2%) of all isolated organisms, of which *K. pneumoniae* accounted for 58.8% (Table 2).

**Table 2 microorganisms-12-01657-t002:** ESBL organisms in CHAMPS sites, Sierra Leone 2019–2022.

Characteristic	N	ESBL ^2^ (*n* = 243) ^1^	Non-ESBL (*n* = 124) ^1^
Isolated organisms	367		
*Klebsiella pneumonia*		143 (94.1%)	9 (5.9%)
*Escherichia coli*		27 (67.5%)	13 (32.5%)
*Enterobacter cloacae*		34 (97.1%)	1 (2.9%)
Other *		39 (27.8%)	101 (72.1%)

^1^ *n* (%); ^2^ Extended-spectrum beta-lactamases. Carbapenem-resistant organisms were identified in 71 (19.3%) of all isolated organisms, of which *K. pneumoniae* accounted for 43.7% (Table 3). * *Burkholderia cepacia* complex, *Klebsiella aerogenes*, *Klebsiella oxytoca*, *Acinetobacter baumannii*, *Staphy-lococcus haemolyticus*, *Staphylococcus epidermidis*, *Enterococcus faecalis*, *Staphylococcus aureus*, *Salmonella species*, *Pseudomonas aeruginosa*, *Serratia marcescens*, *Aeromonas hydrophila.*

**Table 3 microorganisms-12-01657-t003:** CRO organisms (CRO) in CHAMPS sites, Sierra Leone 2019–2022.

Characteristic	N	CRO ^2^ (*n* = 71) ^1^	Non-CRO (*n* = 296) ^1^
Isolated organisms	367		
*Klebsiella pneumonia*		31 (20.4%)	121 (79.6%)
*Escherichia coli*		5 (12.5%)	35 (87.5%)
*Enterobacter cloacae*		9 (25.7%)	26 (74.3%)
Other *		26 (18.6%)	114 (81.4%)

^1^ *n* (%); ^2^ Carbapenem-resistant organisms. * *Burkholderia cepacia* complex, *Klebsiella aerogenes*, *Klebsiella oxytoca*, *Acinetobacter baumannii*, *Staphy-lococcus haemolyticus*, *Staphylococcus epidermidis*, *Enterococcus faecalis*, *Staphylococcus aureus*, *Salmonella species*, *Pseudomonas aeruginosa*, *Serratia marcescens*, *Aeromonas hydrophila*.

The MDR pattern was identified in 258 (70.3%) of all isolated organisms, of which *K. pneumoniae* accounted for 138 (53.5%) (Table 4).

**Table 4 microorganisms-12-01657-t004:** MDR organisms in CHAMPS sites, Sierra Leone 2019–2022.

Characteristic	N	MDR ^2^ (*n* = 258)	Non-MDR (*n* = 109) ^1^
Total isolated organisms	367		
*Klebsiella pneumonia*		138 (90.8%)	14 (9.2%)
*Escherichia coli*		26 (65.0%)	14 (35.0%)
*Enterobacter cloacae*		33 (94.3%)	2 (5.7%)
Other *		61 (43.6%)	79 (56.4%)

^1^ *n* (%), ^2^ Multidrug resistant. * *Burkholderia cepacia* complex, *Klebsiella aerogenes*, *Klebsiella oxytoca*, *Acinetobacter baumannii*, *Staphy-lococcus haemolyticus*, *Staphylococcus epidermidis*, *Enterococcus faecalis*, *Staphylococcus aureus*, *Salmonella species*, *Pseudomonas aeruginosa*, *Serratia marcescens*, *Aeromonas hydrophila*.

Among *K. pneumoniae* isolates, 138/152 (90.8%) presented an MDR pattern. Overall, resistance to penicillin–cephalosporins–aminoglycosides–fluoroquinolones was the most common pattern (61.8%; 94/152), as shown in Figure 2.

Figure 3 summarizes the resistance patterns among *E. coli* isolates. The MDR pattern was observed in 26/40 (65.0%) isolates, with the most common pattern of resistance as follows: penicillins–cephalosporins–aminoglycosides–fluoroquinolones in 31.6% (12/38) of the isolates.

The MDR pattern was observed in 33/35 (94.3%) *E. cloacae* isolates, and the most common pattern of resistance was penicillins–cephalosporins–aminoglycosides–fluoroquin–olones–carbapenems (20/35; 51.7%), as shown in Figure 4.

## 4. Discussion

Our study assessed the ESBL, CRO, and MDR patterns in isolates of *K. pneumoniae*, *E. coli*, and *E. cloacae* among postmortem blood samples taken from stillbirths and under-five children in Sierra Leone over a three-year period. Gram-negative bacteria were the predominantly isolated organisms.

Overall, this study found a high MDR pattern, with *E. cloacae* showing the highest rate (94.3%), followed by *K. pneumoniae* (90.8%). This is comparable to other studies in Ethiopia, which reported MDR in urinary pathogens at 87.4% [24], and in Ghana, at 65% [25]. Multidrug resistance in Enterobacteriaceae is a growing public health concern in Sierra Leone, as in many other parts of the world [2,3,26]. In addition to bloodstream infections, these bacteria are responsible for a variety of other infections, including urinary tract infections, pneumonia, and skin infections. In Sierra Leone, MDR in Enterobacteriaceae is primarily driven by the overuse and misuse of antibiotics, which has led to the emergence and spread of resistant strains [2]. Many healthcare facilities in Sierra Leone lack the resources, personnel, and infrastructure to effectively monitor and control the use of antibiotics, which has contributed to the problem [2,3].

The emergence of extended-spectrum beta-lactamase (ESBL)-producing strains of *Klebsiella pneumoniae* and *Escherichia coli* constitute another poorly understood example of MDR in Enterobacteriaceae in Sierra Leone. ESBLs are enzymes that confer resistance to many commonly used antibiotics, including beta-lactams such as penicillins and cephalosporins. These strains have been associated with high mortality rates and are particularly concerning in healthcare settings such as hospitals. In this study, we found a relatively high prevalence of ESBL with *Enterobacter* spp., *K. pneumonia*, and *E. coli* showing prevalences of 97.1%, 94.1%, and 67.5% respectively. This is similar to studies in South Africa with 83% [27], Nigeria with 83.9% [28], and Ethiopia with 100% [24].

Our estimates are, however, higher compared to recent studies in Portugal, which found 64.4% of isolates [29], and in Iran (76.47%) [30].

ESBL producers are often resistant to the fourth generation cephalosporins in vivo, despite displaying susceptibility in vitro. The presence of ESBL-producing pathogens may result in the overuse of other broad-spectrum antibiotics, leading to inappropriate antimicrobial administration. Antibiotics active against ESBL-producing pathogens include the carbapenems and beta-lactam–beta-lactamase inhibitor combinations, such as piperacillin–tazobactam and cephamycins. In our study, the overall ESBL resistance among the isolates was 66.2%, with *E. cloacae* at 97.1%, followed by *K. pneumoniae* at 94.1% (Table 2). The meta-analysis of ESBL in Enterobacteriaceae among pregnant/postmortem women in Africa, Asia, and Europe showed a prevalence of 45%, 33%, and 2%, respectively [31]. Our study found higher estimates than these. This could be attributed to the ease with which antibiotics are found, dispensed, and self-medicated in our study setting. Another study in the southern region of Sierra Leone corroborates our findings, though the limitation for that study was the small sample size compared to our study (64.3% of ESBL Enterobacteriaceae) [31].

In addition to ESBLs, other mechanisms of resistance have also been identified in Enterobacteriaceae in Sierra Leone, including carbapenemases and plasmid-mediated resistance genes. Carbapenemases are enzymes that confer resistance to carbapenem antibiotics, which are often considered “last resort” antibiotics for treating MDR infections. Plasmid-mediated resistance genes can confer resistance to a wide range of antibiotics and are particularly concerning because they can be easily transferred between different bacterial species. In our study, we found that CRO accounted for 19.3% of the three isolates tested, with the highest resistance shown in *E. cloacae* (25.7%). This finding is comparable to the 25.8% reported in Nigeria [27]. However, our prevalence is much higher than what was reported in Ethiopia, with 2.7% [24], and in Uganda, with 0%. [32].

### Strengths and Limitations

An important strength of this study stems from the fact that access to microbiological culture and sensitivity testing methods is very limited in our setting, so CHAMPS brought novel automated culture and antimicrobial sensitivity testing for samples from sterile sites. However, a limitation of our study is that postmortem blood isolates may not represent the true population prevalence of AMR. In addition, most of the isolated microorganisms may not have been clinically important, and “intermediate resistance” was taken as resistant. To better understand the population prevalence of AMR, a population-wide prevalence study for live cases will be required. Similarly, the isolates in our study may not be attributable as the cause of death for all cases, as we assessed all isolates found in the blood irrespective of their contribution as a cause of illness. However, the presence of these resistance organisms, irrespective of whether they are part of the causal chain to death, remains worrisome.

## 5. Conclusions

This study demonstrated a high prevalence of MDR patterns in *Klebsiella pneumoniae*, *Escherichia coli*, and *Enterobacter cloacae* in postmortem samples collected from stillbirths and children under five years of age. The emergence of ESBL-producing bacteria causing deaths in children under five years of age may be increasing and needs to be monitored. This may reflect the lack of implementation of national antibiotic stewardship policies in Sierra Leone and the widespread access to antimicrobials without prescription. Further prospective studies are required to explore how strategies promoting prudent use of antibiotics and antimicrobial stewardship programs can work hand in hand with the refined infection prevention control.

To effectively address this issue, the government must continue to invest in public health campaigns on the controlled use of antimicrobials; strengthen surveillance and monitoring systems; and increase investment in healthcare infrastructure and resources. It is also important that healthcare providers are educated on antibiotic stewardship and are provided access to laboratory-guided prescription. These measures will help to reduce the emergence and spread of additional resistant strains.

## Figures and Tables

**Figure 1 microorganisms-12-01657-f001:**
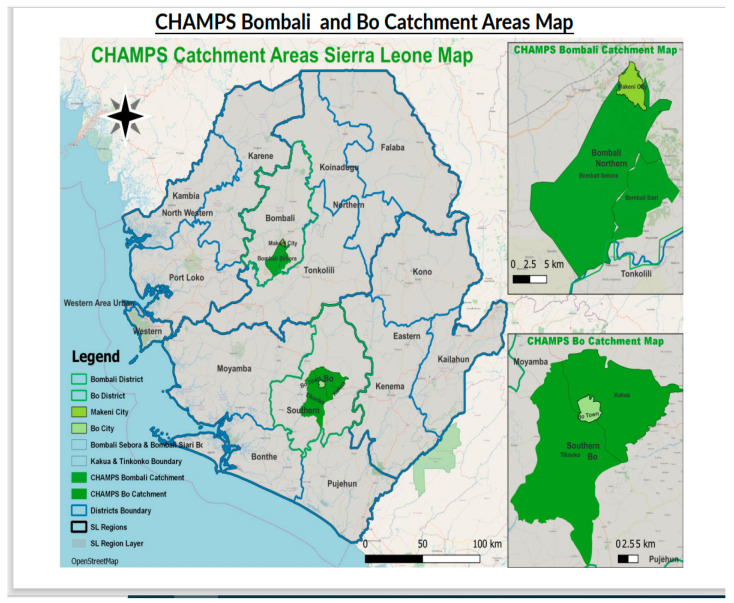
The Bombali and Bo Districts showing CHAMPS study site catchment areas, Sierra Leone.

**Figure 2 microorganisms-12-01657-f002:**
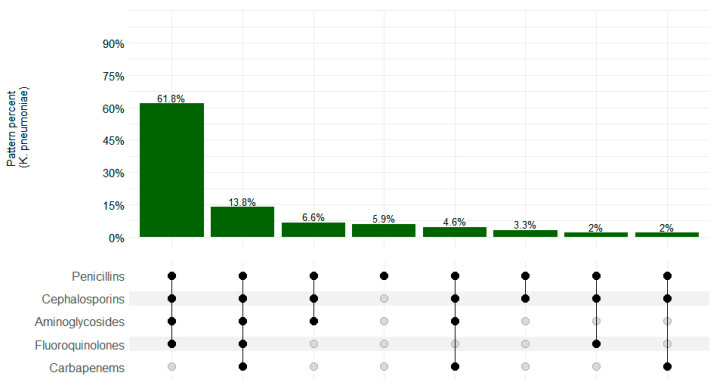
*K. pneumoniae* MDR resistance patterns in CHAMPS sites, Sierra Leone 2019–2022. Black dots depict resistance and gray dots depict sensitivity.

**Figure 3 microorganisms-12-01657-f003:**
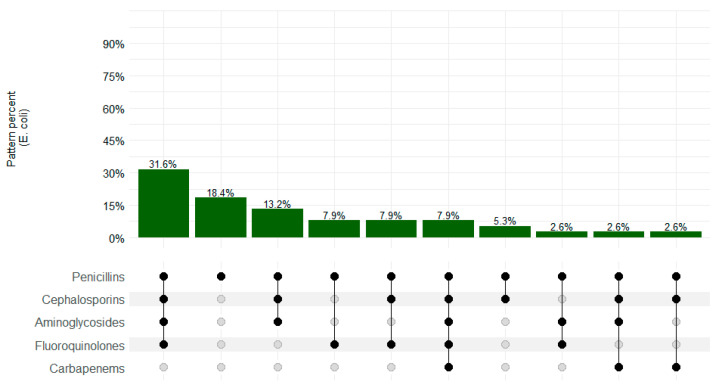
*Escherichia coli* resistance patterns in CHAMPS sites, Sierra Leone 2019–2022. Black dots depict resistance and gray dots depict sensitivity.

**Figure 4 microorganisms-12-01657-f004:**
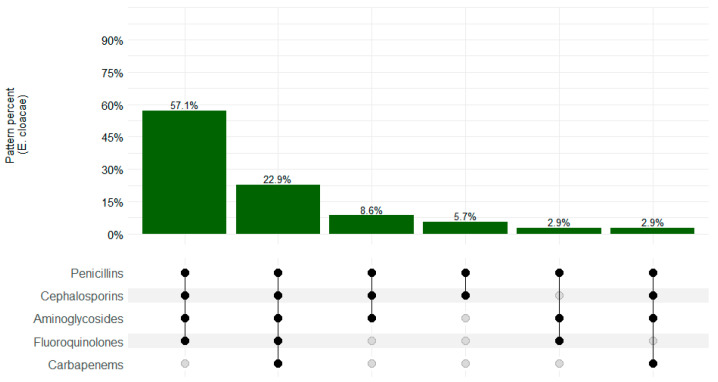
*Enterobacter cloacae* resistance patterns in CHAMPS sites, Sierra Leone 2019–2022. Black dots depict resistance and gray dots depict sensitivity.

## Data Availability

The data presented in this study are openly available in [CHAMPS] at [https://champs.emory.edu/redcap/surveys/?s=PCEERX993Y].

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
