# Peer review of "Prevalence of Antimicrobial Resistance in Klebsiella pneumoniae, Enterobacter cloacae, and Escherichia coli Isolates among Stillbirths and Deceased Under-Five Children in Sierra Leone: Data from the Child Health and Mortality Prevention Surveillance Sites from 2019 to 2022"

_microorganisms, 2024, doi:10.3390/microorganisms12081657_

Round 1

Reviewer 1 Report

Comments and Suggestions for Authors

The paper investigates the prevalence of antimicrobial resistance in Klebsiella pneumoniae, Enterobacter cloacae, and Escherichia coli isolates from stillbirths and deceased under-five children in Sierra Leone.

Line 69> add a space between "ESBL" and "and"

Line 88> please replace pattern with its plural patterns

Line 69> replace carbapenemase producing organisms with carbapenem resistant organisms (CRO)

Line 136> why is piperacillin-tazobactam included in cephalosporin class? Please revise the MDR definition and re-run the analysis involving MDR. That involves probably changes in the numbers in all tables and figures.

Table 2> please specify whta bacterial species are included in the other category.

Table 2>you added a superscript 1 explained below as n(%). The table legend should be written below the last black line of the table. The superscrip 1 is absent in the ESBL column.

Table 2> please add in the legend explanation of ESBL (as in Table 3 for CRO). See above comment for legend placement.

Figures 2-4. It should be interesting to see also what ppercent of isolates were susceptible to all antibiotics. Please add this column (if such isolates exist).

Reviewer 2 Report

Comments and Suggestions for Authors

Comments and suggestions

Summary section:

1. Specify the gaps in knowledge regarding the susceptibility patterns of these bacteria.

2. What is the criterion for establishing the sampling within 24 hours post mortem.

3. There were 367 samples in total? Were some excluded?

4. The conclusions should be more precise and focused on the objectives of the study.

Introduction section:

5. Specify the main pathogens that are considered a global threat according to current reports.

6. Change reference 1 to a more pertinent one.

7. Support the paragraph in references 5 and 6 with what is mentioned in the most current clinical practice guidelines.

8. In the third introductory paragraph: Is there any support or treatment guide from the local health authority that supports the use of cephalosporins and carbapenems in these patients?

9. State the main bacterial pathogens that cause neonatal sepsis

10. State some mechanisms by which these bacteria "escape" the effect of antibiotics.

11. References 10 and 11 fail to support the paragraph in which they are indicated

12. Eliminate the term exclusively from the last paragraph of the introduction

Methods section

13. Place the study population in the objective of the manuscript

Methods section:

14. Why was the blood sample taken within the first 24 hours?

15. Justify why intermediate resistance was considered resistant and not as an individual category?

Discussion section:

16. Restructure the first 2 paragraphs based on the results of the study.

Conclusion section

17. The conclusion should be restructured only based on the results and considering the limitations. For example, if in limitations it clearly mentions that this study, due to its methodology and the number of samples, may not represent the true population prevalence, conclusions cannot be drawn that reflect the lack of implementation of national policies, etc.

Comments on the Quality of English Language

 Minor editing of English language required

Reviewer 3 Report

Comments and Suggestions for Authors

Thank you for inviting me to review this manuscript. It is interesting and well-written. I have some comments that could be of use:

·      Line 29: better say babies, children, or something like that instead of ‘cases’

·      Line 53: Better cite here the corresponding study that defines what MDR, XDR, and PDR are (Magiorakos 2012)

·      Line 69: insert space

·      Remove double spaces (such as in line 80)

·      Line 128: Any organism showing intermediate resistance was taken as resistant. è Why? Usually, intermediate means ‘sensitive in higher exposure’

·      Methods section: information regarding the period the inclusion and the exclusion criteria should be provided

·       Line 150: since the data do not refer to the whole of 2019 or 2022, the authors should also provide the months again

·      Line 160: You probably mean ‘carbapenem-resistant’. This needs to be corrected throughout the text (e.g., in Table 2)

·      Tables: It would be better to define all abbreviations used in the tables in a separate footnote for each table

·      Do the authors have any data on Gram-positive pathogens that could relate to the clinical context? Vancomycin-resistant Enterococcus could be such an example. Notably, 38% of the isolated microorganisms are not described here. It would be helpful to provide data for these microorganisms as well, even in the form of a supplementary table

·      Another important limitation of the study, beyond the fact that the isolated microorganisms may not have been clinically important, as the authors have mentioned, is that the previous treatment of the children has not been mentioned. This would be an interesting information to add and possibly to correlate with the presence of antimicrobial resistance

Comments on the Quality of English Language

Minor changes

Round 2

Reviewer 2 Report

Comments and Suggestions for Authors

The authors have responded fairly satisfactorily to the comments and suggestions raised. The high percentage of similarity of the manuscript is due to the fact that it was published as a pre-print.

Comments on the Quality of English Language

Minor editing of English language required

Author Response

Dear Editor

Reviewer 3 Report

Comments and Suggestions for Authors

The manuscript has been improved. However, I disagree with assigning 'intermediate' as 'resistant' strains. Maybe this should be expanded, or added as a limitation.

Comments on the Quality of English Language

Minor

Author Response

Dear Editor,

Please see attached
